# Partial Implant Rehabilitations in the Posterior Regions of the Jaws Supported by Short Dental Implants (7.0 mm): A 7-Year Clinical and 5-Year Radiographical Prospective Study

**DOI:** 10.3390/jcm13061549

**Published:** 2024-03-08

**Authors:** Miguel de Araújo Nobre, Carolina Antunes, Armando Lopes, Ana Ferro, Mariana Nunes, Miguel Gouveia, Francisco Azevedo Coutinho, Francisco Salvado

**Affiliations:** 1Research, Development and Education Department, Malo Clinic, Avenida dos Combatentes, 43, Level 11, 1600-042 Lisboa, Portugal; cantunes@maloclinics.com; 2Clínica Universitária de Estomatologia, Faculdade de Medicina, Universidade de Lisboa, 1649-028 Lisboa, Portugal; franciscoac@outlook.com; 3Oral Surgery Department, Malo Clinic, Avenida dos Combatentes, 43, Level 9, 1600-042 Lisboa, Portugal; alopes@maloclinics.com (A.L.); aferro@maloclinics.com (A.F.); mnunes@maloclinics.com (M.N.); mgouveia@maloclinics.com (M.G.)

**Keywords:** dental implants, immediate loading, edentulous jaws, dental prosthesis, implant-supported, follow-up study

## Abstract

**Background**: Short implants have been used in the restoration of edentulous jaws in the past several years. However, some studies have suggested that short implants are less successful than standard implants. The aim of this study is to investigate the outcome of short implants placed in the posterior maxilla or mandible following one-stage or immediate-function protocols with a follow-up of 7 years (clinically) and 5 years (radiographically). **Methods**: This study included 127 patients rehabilitated with 217 implants measuring 7 mm and supporting 157 fixed prostheses in the posterior segments of both jaws. Final abutments were delivered at the surgery stage and were loaded after 4 months in 116 patients (199 implants). The primary outcome measure was implant survival measured through life tables. Secondary outcome measures were marginal bone loss and the incidence of biological and mechanical complications at the patient level and implant level (evaluated through descriptive statistics). **Results**: Twenty-four patients (18.9%) with 45 implants (20.7%) were lost to the follow-up. In total, 32 implants failed (14.8%) in 22 patients (17.3%), resulting in a cumulative survival rate at 7 years of 81.2% for 7 mm implants in the rehabilitation of the posterior regions of the maxilla and mandible. The average (standard deviation) marginal bone loss was 1.47 mm (0.99 mm) at 5 years. The incidence rate of biological complications was 12.6% and 10.6% at the patient and implant levels, respectively. The incidence rate of mechanical complications was 21.3% for patients and 16.1% for implants. A higher failure rate was registered in smokers and in implant arrangements with a sequence of three fixtures in proximity. **Conclusions**: Within the limitations of this study, it can be concluded that the placement of 7 mm long implants for the partial implant-supported rehabilitation of atrophic posterior jaws is possible in the long term, judging by the survival rate and stable average marginal bone loss. Nevertheless, strict case selection should be performed, especially in smokers and with implant arrangements that provide a minimum of one unit in inter-implant distance.

## 1. Introduction

Since the beginning of the millennium, short implants have been used in the restoration of edentulous jaws [1,2]. However, some studies have suggested that short implants are less successful than standard implants are, yielding lower survival rates [3,4]. Short implants present several advantages, as they are an option for the rehabilitation of extremely resorbed posterior jaws, allowing us to avoid other techniques such as bone augmentation surgeries, which are more invasive, with higher morbidity rates and higher costs [4]. However, short implants also present some disadvantages, such as less bone to implant contact and a higher crown/implant ratio, when compared with standard implants [4].

The definitions of short implant differ between authors. While some authors consider short implants to be ≤10 mm in length [5,6,7], others consider short implants to measure ≤8.5 mm [8,9] or ≤8 mm [10,11].

In addition to length, micro-implant design (the implant’s surface) also plays a potentially important role in implant survival. Evidence suggests that oxidized surfaces lead to lower rates of implant failure compared with machined surfaces [5,12]. In short, implants placed in the maxilla, anodically oxidized surface implants (TiUnite; Nobel Biocare AB, Gothemburg, Sweden), registered a 7.1% failure rate, representing a 13.3% decrease compared with that of machined surface implants [13]. Additionally, the placement of implants in posterior areas is more challenging when compared with that in anterior regions, due to the lower bone quantity, decreased bone density, and increased occlusal loading in these sites [14].

In recent years, several studies have estimated the survival rate of short implants (≤10 mm in length): retrospective studies with follow-ups between 1 year and 10 years registering short implant cumulative survival rates ranging between 88.9% and 98.3% [9,10,14,15,16,17,18] with different study designs, sample characteristics, and follow-up durations. However, lower survival rates were registered for 7 mm implants [15,16], dropping dramatically to 81.5% in one of the studies [15]. Considering immediate function, previous studies revealed a cumulative survival rate of 95.6% and 95.9% at 5 years and a mean of 5.2 years, respectively [9,17], while a previous study from our group registered a survival rate of 93.7% at 3 years for implants measuring 7 mm in length [14]. The protocol for the placement of short implants, compared with other techniques for the rehabilitation of the posterior jaws that are more invasive, makes a difference in terms of predictability, patient compliance, and implant survival [19,20].

Given the low number of studies documenting longer outcomes for implants of short lengths (7 mm) with or without immediate function, and the different survival outcomes registered, it is deemed necessary to further document single-unit and partial rehabilitations using these implants. The purpose of this study was to investigate implant survival, marginal bone loss, and the mechanical and biological complications of short implants placed in the posterior maxilla or mandible following an immediate-function protocol, assuming the hypothesis about posterior regions with a predictable positive outcome.

## 2. Materials and Methods

This article was written following the STROBE (Strengthening the Reporting of Observational studies in Epidemiology) guidelines [21]. This prospective case series comprises data collected on implants measuring 7 mm in length (Figure 1), inserted in the maxillae and mandibles to rehabilitate single- and partially edentulous conditions. The study was conducted at a private rehabilitation center and was approved by an institutional clinical/human experimentation panel (Ethical Committee for Health, Lisbon, Portugal, authorization no. 004/2009), following the Helsinki Declaration of 1964, as revised in 2013. All patients provided informed consent to participate in the study. Data reporting was chosen to be performed at the stages of short-, medium- and long-term outcomes (1 year, 3 years, and more than 5 years of follow-up). 

The patients were included in the study provided that they needed implant-supported restorations and had a severely atrophied jaw, including a bone height between 5 and 7 mm in the maxilla below the maxillary sinus and of 7 mm in the mandible above the mandibular canal. The following exclusion criteria were applied: patients hindered from providing informed consent for participating, patients presenting immunodeficient pathology, bruxism, or emotional instability, undergoing maxillary or mandibular radiation therapy, undergoing active chemotherapy, or in need of bone grafting procedure. 

The surgical procedure was previously described in detail in a previous report with a 1-year follow-up [22]. In brief, surgery was performed under local anesthesia and the insertion of the implants followed standard procedures [22], with the following modifications: incisions performed on the crest’s palatal aspect with parallel discharges towards the vestibulum to allow maximum tissue repositioning of the vestibular aspect of the flap (no connective tissue grafts were used); the drilling sequence modified via under-preparation for maximum implant insertion torque. Considering the implant macro-design, the implant platform was aimed to be placed flush to the bone crest and bicortical anchorage was established whenever possible. All implants were placed with insertion torques ranging from 32 to 50 Ncm. 

The abutment selection was based on the type of rehabilitation: for fixed partial prostheses, multiunit abutments (Nobel Biocare AB) were chosen, while for single teeth, healing abutments (Nobel Biocare AB) were used. The soft tissues were readapted and repositioned with 4-0 non-resorbable sutures (Braun Silkam non-absorbable 4-0, Aesculap, Tuttlingen, Germany). Healing caps were connected to the definitive abutments. In 11 patients (18 implants), a provisional acrylic resin crown or prosthesis was manufactured and attached to the implants or abutments (immediate temporary abutments, Nobel Biocare AB, were used for cement-retained crowns) on the day of surgery and were left in infra-occlusion. Four months after, provisional prostheses were manufactured and connected to the implants. Final prostheses were provided typically at 6 months, consisting of ceramic crowns or ceramic/metal-ceramic prostheses. 

All patients were evaluated clinically up to 7 years and radiographically at 1-, 3- and 5 years. The pretreatment orthopantomography is presented in Figure 2.

Primary outcome measures were prosthetic and implant survival. As secondary outcome measures, marginal bone level changes and biological and mechanical complications were evaluated. Prosthetic survival was based on function; the necessity to replace the definitive prosthesis was considered a failure. An implant was classified as attaining survival based on the fulfillment of its purported function as support for reconstruction and stability when tested manually (the removal of prosthesis and implants was tested manually). Implants that did not meet the survival criteria and implants that were removed were considered failures.

Periapical radiographs were taken using the parallel technique with a film holder (Super-bite, Hawe-Neos, Bioggio, Switzerland) and an aiming device on the day of implant insertion and 5 years post-surgery. An outcome assessor examined all periapical radiographs. Each periapical radiograph was scanned at 300 dpi with a scanner (HP Scanjet 4890, HP Portugal, Paço de Arcos, Portugal), and the marginal bone level was measured with an image analysis software (Image J version 1.40 g for Windows, National Institutes of Health, Stapleton, NY, USA). The reference point for the reading was the implant platform (the horizontal interface between the implant and the abutment), and marginal bone loss was defined as the difference in the marginal bone level relative to the bone level at the time of surgery as seen on the periapical radiograph. The radiographs were accepted or rejected for evaluation based on the clarity of the implant threads; a clear thread guaranteed both sharpness and the orthogonal direction of the radiographic beam towards the implant axis. The dimensions on the radiographs were calibrated using the distance between implant threads as a reference.

The following complication parameters were assessed: fracture or loosening of mechanical and prosthetic components (mechanical complications), peri-implant pathology, soft tissue inflammation, fistula formation, pain, nerve damage (biological complications), and aesthetic complains from the patient or dentist (aesthetic complications). 

Descriptive statistics were used to classify the variables of interest, namely marginal bone loss (average and standard deviation) and the incidence of complications. Prosthetic and implant survival were evaluated through life tables. In order to clarify the main factors that influenced the survival rate, several comparisons were performed, including those regarding sex (male vs. female), health status (healthy vs. presence of systemic condition), smoking habits (smokers vs. non-smokers), number of implants in sequence (one or two implants vs. three implants), and the location of the rehabilitation (mandible vs. maxilla). The overall inter-examiner reliability was estimated using a weighted average of the pairwise inter-examiner reliability estimates. Inter-examiner reliability results for weighted kappa scores were 0.83.

## 3. Results

### 3.1. Sample Selection

The study included 127 patients (96 females and 31 males), with an age range of 23–78 years (mean = 53.1 years, standard deviation = 10.1 years). In total, 23 patients (12.6%) were smokers (average of 15 cigarettes per day), and 37 patients (29.1%) presented a systemic condition: a cardiovascular condition (*n* = 22 patients; 17.3%) or thyroid condition (*n* = 4 patients; 3.1%); diabetes mellitus (*n* = 5 patients; 3.9%); rheumatologic condition (*n* = 10 patients; 7.9%); hepatitis (*n* = 1 patient; 0.8%); a previous oncologic condition (*n* = 3 patients; 2.4%); an autoimmune condition (*n* = 1; 0.8%). Nine patients (7.1%) presented more than one condition. Implant surgery and prosthetic placement were performed by the same team. The first implant was placed in July 2005 and the last was placed in July 2009; these were followed for 7 years clinically and 5 years radiographically. The patients were referred to the clinic as candidates for single or partial rehabilitation in the posterior regions of both jaws.

### 3.2. Implants

In the present study, short implants measuring 7 mm in length were used. In total, 217 short implants (NobelSpeedy Shorty, Nobel Biocare AB) were placed (21 in the maxilla and 196 in the mandible), out of the total of 157 rehabilitations (20 in the maxilla and 137 in the mandible), with 141 single teeth (16 in the maxilla and 125 in the mandible) and 16 partially fixed prostheses (4 in the maxilla and 12 in the mandible). A case illustration is presented in Figure 3, Figure 4 and Figure 5.

The number of implants per patient and their position are indicated in Table 1, with most patients having one short implant, and with the implants inserted in the positions between the first premolar and the second molar (Table 1).

The implant was 4 mm in diameter and 7 mm in length, and had an external connection, anodically oxidized surface and collar (TiUnite^TM^, Nobel Biocare), slightly larger edges on threads, grooves added to the threads, and a tapered apex (Figure 1).

Briefly, 23 patients (representing 28 implants) were smokers, and 45 patients had implants placed in periodontally compromised sites (representing 59 implants). 

### 3.3. Implant Survival Rate

At 7 years, 24 patients (18.9%) representing 45 implants (20.7%) were lost to the follow-up (Table 2).

Briefly, two prostheses failed (1.6%) in two patients, rendering the prosthetic survival rate 98.7%. In total, 32 implants failed (14.8%) in 22 patients (17.3%), resulting in a cumulative survival rate at 7 years of 81.2% for short implants in the rehabilitation of the posterior maxilla and mandible, (Table 3, Figure 6). The implants placed in smokers showed a 4.5-fold increase in the failure rate (smokers: 10 failures and 17 survivals; non-smokers: 22 failures and 168 survivals).

The implant cumulative survival at 7 years considering different factors is depicted in Table 4. Higher failure rates were registered for males and smokers in a three-implant sequence and in the maxilla.

### 3.4. Marginal Bone Loss

Marginal bone loss was evaluated at the 5-year follow-up (Table 5). There were readable radiographs for 107 implants in 76 patients, out of a total of 90 patients that were followed up for 5 years (164 implants). The mean (standard deviation) marginal bone loss measured at 5 years was 1.47 mm (0.99 mm). 

### 3.5. Biological Complications

Biological complications (Table 6) were present in 16 patients and 23 implants (12.6% and 10.6%, respectively) that consisted in peri-implant pathology (probing pocket depths measuring ≥5 mm, marginal bone loss, bleeding on probing, and suppuration). These complications were addressed via non-surgical treatment (the removal of bacterial plaque and irrigation with 0.2% chlorhexidine gel), and these treatments were successful in 10 implants. Two implants were lost, and the remaining implants maintained probing pocket depths measuring ≥5 mm.

### 3.6. Mechanical Complications

Mechanical complications (Table 7) occurred in 35 implants (16.1%) and 27 patients (21.3%). These complications included crown fractures (*n* = 8; 22.9%) and the loosening of prosthetic screws (*n* = 12; 34.3%) and abutment screws (*n* = 15; 42.9%). From the eight crown fractures, six were in provisional acrylic prostheses and two were in definitive ceramic prostheses. Five fractures were resolved by replacing the prostheses (three provisional prostheses and two definitive prostheses), and three prostheses were mended in the laboratory; the loosening of abutments and prosthetic screws was resolved by retightening and adjusting the occlusion, and in 12 patients, a nightguard was provided.

## 4. Discussion

The outcomes of the present study demonstrated that using implants measuring 7 mm in length with an oxidized surface in challenging areas (atrophic posterior regions of the jaws) is viable but should be considered with caution, especially in smokers and for implant arrangements of more than two units. Several systematic reviews and meta-analyses reported survival rates for short implants to be between 88.1 and 97.89% [3,5,11,12,13,18,23,24]. However, these studies have extensive follow-up ranges, and some have short follow-ups. When considering studies with at least the same follow-up range as the present study, cumulative survival rates ranged from 80.3% [25] to 98.3% [10]. The cumulative implant survival rate registered in the present study warrants discussion concerning the differences compared with other studies. Considering the number of implants in sequence, a study from Lai et al. [10] reported a cumulative survival rate of 98.3% for short implants (≤8 mm), with a mean period of 7.22 years. However, this study only used single-tooth rehabilitations, which, in our study, led to significantly higher implant survival rates when compared with those for implant arrangements including more than two implants in sequence. A previous study identified a potential increase in marginal bone loss for smaller inter-implant distances. Tarnow et al. [26], in a study evaluating the effect of inter-implant distance, registered a two-fold increase in marginal bone loss for an inter-implant distance ≤3 mm compared with that for a distance of >3 mm. It is important to highlight that in posterior atrophic regions submitted to high occlusal loads, an increase in the number of short implants for improved occlusal support did not correlate with better outcomes in our study. 

Regarding the implant location (maxilla or mandible), Mendonça et al. [16] reported a cumulative survival rate of 95.45% for short implants (≤10 mm) during a mean period of 9.7 years. Nevertheless, 393 (86.8%) of the 453 implants were placed in the mandible and heavy bruxers were excluded. The present study revealed that implants placed in the mandible showed a higher survival rate (83.1%) when compared with implants placed in the maxilla (64%) while including patients that developed bruxing habits. These results may be explained through several conditions present when placing implants in the maxilla, such as insufficient vertical bone volume, poor bone quality, reduced interarch space, and limited visibility [27]. 

Concerning the distribution of implants according to sex and the time of follow-up, a previous retrospective study by Malo et al. [28] reported a cumulative survival rate of 96.2% at 5 years, for 7 mm implants with immediate function. This study had only 50 patients (38.2%) who overcame the 5-year follow-up period. The results of the present study demonstrated that complications and implant failure were aggravated from the fifth year of follow-up onwards, and thus more studies are necessary to accurately measure the long-term outcome of short implants placed in posterior jaws. In addition, most of the sample was composed of women (71.3%) in the study by Maló et al. [28], who in the present study were registered to have a higher implant survival rate compared with men (a 20% decreased failure rate in women), a result that is in agreement with that of a previous systematic review [29]. This result might also have been influenced by the distribution of smoking habits, with only 8.8% of women being smokers, compared with 25.5% of men. Additionally, men’s physiognomy is commonly more robust, with stronger masseters and consequently higher occlusal loadings [29]. Finally, men have usually poorer levels of oral hygiene compared with women [30,31].

Other studies registered similar outcomes on the use of short implants: Stellingsma et al. [1], in a retrospective study, reported a cumulative survival rate of 88% for short implants (≤10 mm) in edentulous patients, with a mean follow-up period of 77 months, for implants placed exclusively in the mandibles, and for the majority of women (82%). Brodcard et al. [25], in a prospective study, reported a cumulative survival rate of 80.3% for implants measuring < 8 mm at 7 years. 

There are other treatment alternatives available to rehabilitate partially edentulous jaws with, such as standard implants in grafted bone with open or closed sinus elevation with or without regeneration, nerve transposition (mandible), and alveolar distraction osteogenesis. Several studies compared the performance between short implants inserted in residual bone and standard implants inserted in augmented bone with differing results [24,32,33,34,35,36]. Terheyden et al. [24] registered no significant difference in implant survival at 8 years of follow-up using short implants (≤7 mm) or standard implants in augmented bone; however, less complications and decreased marginal bone loss occurred in short implants. Sáenz-Ravello et al. [36] reported a decreased risk of implant failure, marginal bone loss, and biological complications for short implants in residual bone in the posterior mandible. Buser et al. [32] and Hallman et al. [33], in 5-year prospective studies, both registered cumulative survival rates of 86% and 100% for implants placed in augmented bone, and in patients subjected to maxillary sinus floor augmentation, respectively. Wen et al. [37] reported a 64.1% implant survival rate at 8 years for implants inserted in patients with lateral sinus floor elevation. Khoury at al. [38] performed the insertion of standard implants in sinus floor elevation using a two-layer grafting technique, registering a 99.5% survival rate at 10 years. Jensen et al. [35] performed inferior alveolar nerve transposition followed by the immediate placement of implants in the atrophic posterior mandibular alveolar ridge. Although reporting implant stability between 12 and 46 months, the authors indicated that the surgical technique needed to follow strict patient selection, with patients being fully aware of the possibility of nerve paresthesia. Enislidis et al. [34] evaluated the complications of alveolar distraction osteogenesis and implant placement in the partially edentulous mandible. After a mean follow-up period of 35.7 months, complications associated with this technique affected 75.7% of the patients; for 11 implants, secondary grafting procedures were necessary, registering an implant survival rate of 95.7%. All the studies described had shorter follow-ups compared with that of the present study using short implants measuring 7 mm, which did not allow us to fully assess the outcome of these alternative techniques in the long term. Nevertheless, it becomes clear that the rehabilitation of posterior atrophic regions is challenging irrespective of the treatment alternative. Additionally, all these treatment alternatives require extensive healing periods and several surgical interventions before loading a prosthesis. This represents a disadvantage compared with the use of short implants in one-stage or immediate loading that allow for only one surgical intervention to be used and decreased times between implant insertion and prosthetic loading.

One of the potential impacts on the outcome is the available width of keratinized mucosa around the implants, a topic that has warranted considerable discussion. On one hand, previous investigations reported that the presence or absence of keratinized mucosa did not significantly affect implant survival, resulting in at least a non-direct impact on the outcome [39]. However, a previous systematic review and meta-analysis evaluating the influence of the width of keratinized mucosa on soft and hard tissue outcomes concluded that the lack of keratinized mucosa negatively affected the prevalence of peri-implant pathology, plaque accumulation, soft tissue inflammation, mucosal recession, and marginal bone loss [40], which could impact implant survival and success at least indirectly. Despite the lack of long-term studies, new techniques for keratinized mucosa width augmentation have been described in the recent literature, including a xenogeneic collagen matrix, keratinized mucosa shiftings, or mesh-free gingival grafts [41,42,43,44,45,46]. A long-term clinical prospective study on 13 patients investigating the efficacy of the xenografic porcine collagen matrix in the augmentation of keratinized mucosa revealed stable results in maintaining a width of >2 mm at 10 years of follow-up [41]. Nevertheless, more studies are necessary to determine the impact of these techniques on the long-term outcome.

One of the main findings of the present study suggests that 7 mm implants placed in the posterior jaws might be challenging for smokers in the long term, with a cumulative survival rate of 43.5%, representing 4.5-fold increased odds of implant failure compared with that in non-smokers. Considering the academic hypothesis of excluding smokers from our sample, the cumulative survival rate at 7 years would have been 88.4%, which is a more approximate value of what would have been expected considering previous studies. Brodcard et al. [25] reported a cumulative survival rate of 80.3% and concluded that smoking did not affect the implants negatively; however, the cumulative implant survival was even lower, and the smokers included in the study were mainly light smokers.

In our study, the number of implants in sequence (one or two implants vs. more than two units) appeared to also exert a deleterious influence on implant survival, with more than two units having a registered decrease of 34.7%. Similarly to the present study, a study with short implants (<8 mm) registered a 100% cumulative survival rate for single-tooth rehabilitations, while for rehabilitations of more than one tooth it was 89% at 7 years [25]. 

In the present study, we reported a marginal bone loss of 1.47 mm. Lai et al. [10] reported this to be 0.63 mm (SD 0.68 mm) after a mean period of 7.22 years, and Mendonça et al. [16] reported a marginal bone loss of 1.2 mm, at a mean period of 9.7 years; both studies included smokers. These studies reported lower marginal bone loss compared with that in the present study, which had a similar follow-up period. However, the present study followed a one-stage surgery protocol, which may have affected 0this result, since the baseline is accounted for from implant insertion, while two-stage protocols consider the baseline to be only at prosthesis placement. Terheyden et al. [24], in a systematic review and meta-analysis, registered a range between 1.22 mm and 1.72 mm at the 5-year follow-up, which reflect the results registered in our study. Nevertheless, the proportion of marginal bone loss has different impacts in standard or short implants; certain marginal bone loss in a short implant may represent almost 1/3 of the implant’s bone coverage (the bone level of the implant’s middle third), while in a standard implant, the same bone loss could represent only 1/5 of this (the bone level of the implant’s coronal third).

Concerning the complications, in the present study, we reported rates of biological complications of 12.6% and 10.6% at patient and implant levels, respectively, a rate of 21.3% for patients, and rate of 16.1% for implants regarding mechanical complications. Lai et al. [10] registered lower rates of complications: 7.8% and 12.6% for biological and mechanical complications, respectively, at the implant level. However, they only performed single-tooth rehabilitations and excluded patients with uncontrolled systemic conditions. 

The rate of patients lost to follow-up of below 20% accounts for good internal validity. Limitations concern the study design (case series), the lack of a control group, the fact that this study was single-centered, and the absence of any registration of bacterial plaque indexes. Future research should focus on the long-term outcome (10 years) and use different implant arrangements to investigate the effect on implant survival and marginal bone loss. 

## 5. Conclusions

The present study concludes that the placement of 7 mm long implants placed in one stage or in immediate function for the support of implant-supported prosthesis is an alternative for the partial rehabilitation of the posterior jaws. The 7-year clinical follow-up registered an acceptable cumulative survival rate, while the 5-year marginal bone loss average was low. Nevertheless, careful selection is mandatory considering the inclusion of patients with smoking habits or more than two implants in sequence. 

## Figures and Tables

**Figure 1 jcm-13-01549-f001:**
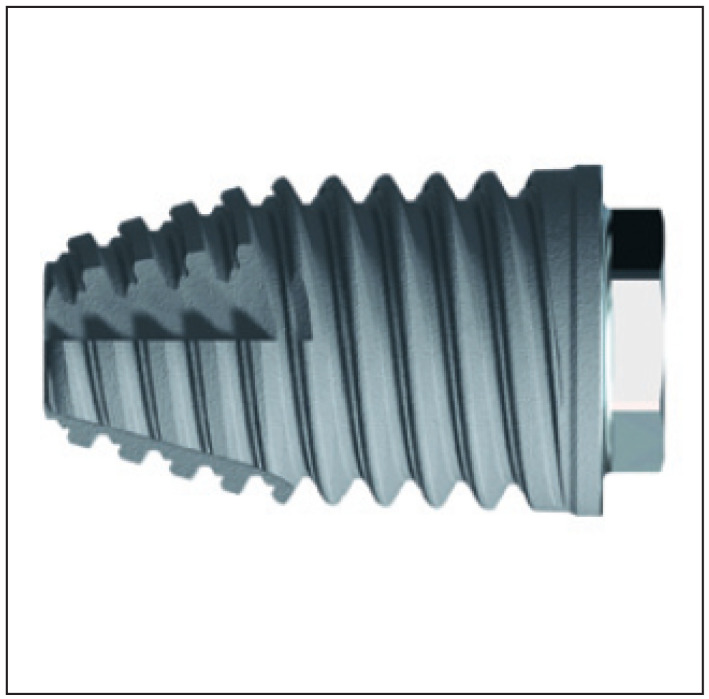
NobelSpeedy Shorty implant (7.0 mm length; 4.0 mm diameter) used in the present study.

**Figure 2 jcm-13-01549-f002:**
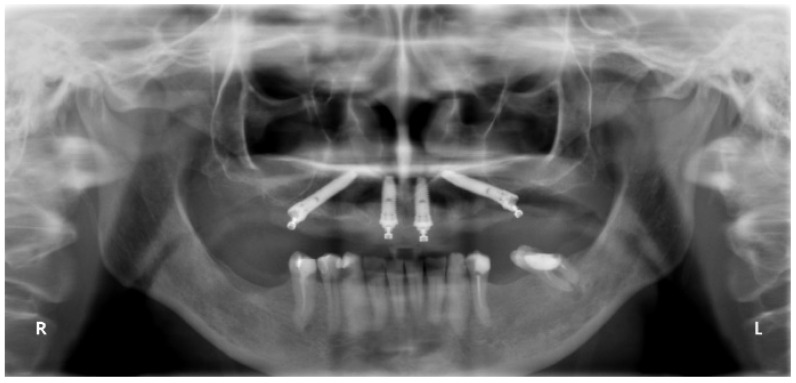
Pretreatment orthopantomography.

**Figure 3 jcm-13-01549-f003:**
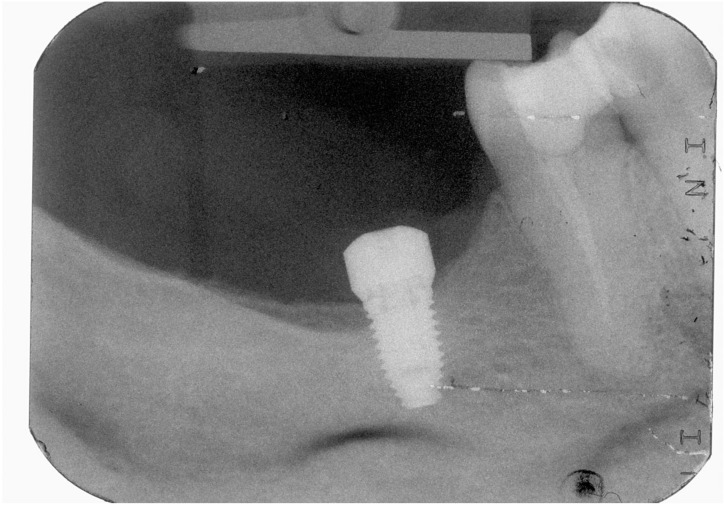
Follow-up periapical radiograph after the insertion of the short implant (4th quadrant).

**Figure 4 jcm-13-01549-f004:**
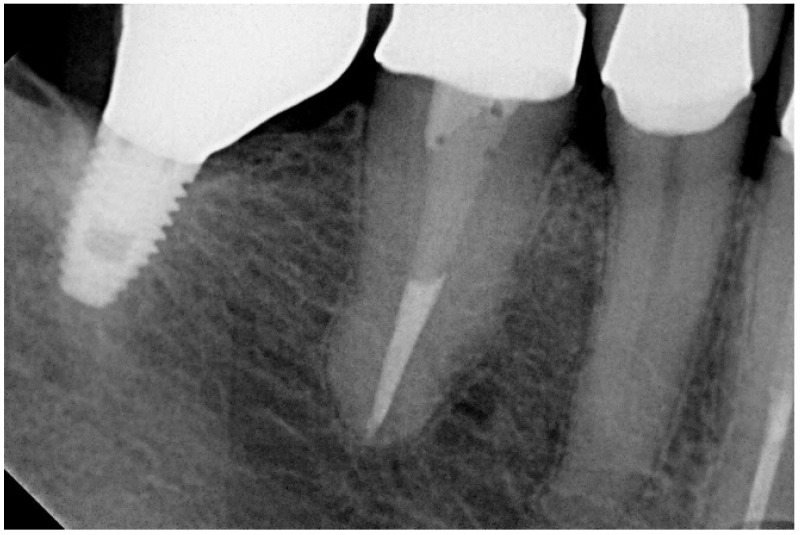
Follow-up periapical radiograph illustrative of the 5-year follow-up of the short implant (4th quadrant).

**Figure 5 jcm-13-01549-f005:**
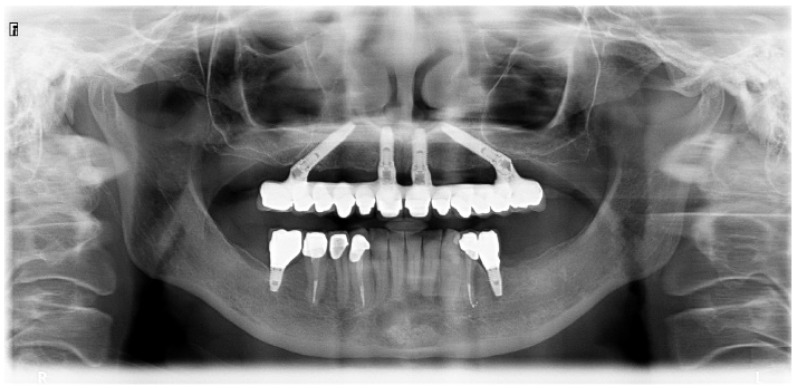
Seven-year follow-up orthopantomograph with the definitive prosthesis connected to a 7 mm implant (4th quadrant).

**Figure 6 jcm-13-01549-f006:**
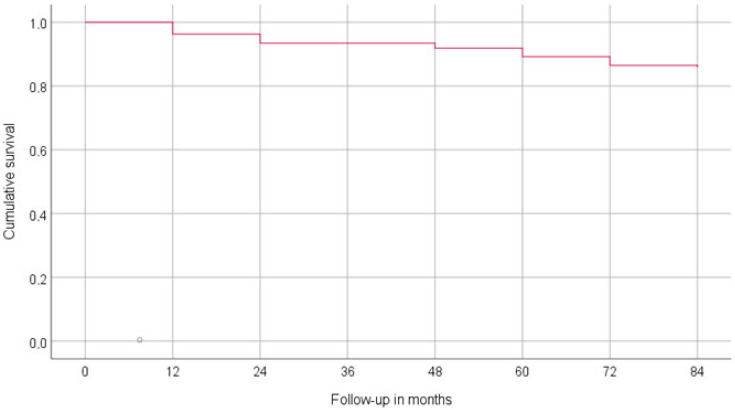
Survival plot illustrating the cumulative survival rate of short implants during the 7-year follow-up.

**Table 1 jcm-13-01549-t001:** Distribution of short implants by number, * position, and status (lost implants).

**Maxilla**									
Position	17	16	15	14	24	25	26	27	*n*
Number of implants	0	6	2	2	0	0	10	1	21 (100%)
Lost implants	0	1	1	2	0	0	3	0	7 (33.3%)
**Mandible**									
Position	47	46	45	44	34	35	36	37	*n*
Number of implants	24	43	22	3	4	28	53	19	196 (100%)
Lost implants	2	6	4	0	0	4	7	2	25 (12.8%)

* Number of short implants per patient: 1 implant (*n* = 73 patients); 2 implants (*n* = 33 patients); 3 implants (*n* = 9 patients); 4 implants (*n* = 7 patients); 5 implants (*n* = 2 patients); 6 implants (*n* = 2 patients).

**Table 2 jcm-13-01549-t002:** Number of patients lost to the follow-up during the present study.

Follow-Up	Number of Patients	Number Lost to Follow-Up	Reasons
0–1 year	127	2	2 patients became unreachable
1–2 years	125	4	4 patients became unreachable
2–3 years	121	4	4 patients became unreachable
3–4 years	117	4	3 patients became unreachable; 1 patient was followed up in a different clinic
4–5 years	113	3	3 patients became unreachable
5–6 years	110	3	3 patients became unreachable
6–7 years	107	4	1 patient deceased; 2 patients became unreachable; 1 patient moved abroad

**Table 3 jcm-13-01549-t003:** Cumulative survival rate of short implants placed in the posterior areas.

Time (Years)	No. of Implants	No. of Failures	Lost to Follow-Up	Survival Rate	Cumulative Survival Rate
0	217	8	3	96.3%	96.3%
1	206	6	8	97.0%	93.4%
2	192	0	7	100%	93.4%
3	185	3	7	98.3%	91.9%
4	175	5	6	97.1%	89.2%
5	164	5	4	96.9%	86.5%
6	155	1	10	99.3%	85.9%
7	144	4	32	94.6%	81.2%

**Table 4 jcm-13-01549-t004:** Cumulative survival rate of short implants by factor at 7-year follow-up.

Factors		Cumulative Survival Rate
Sex	Male	63.7%
	Female	85.6%
Smoking	Smoker	43.5%
	Non-smoker	85.9%
Implant arrangement	One–Two implants	84.1%
	Three implants in sequence	49.4%
Rehabilitation location	Mandible	83.1%
	Maxilla	64.0%

**Table 5 jcm-13-01549-t005:** Marginal bone loss for short implants at the 5-year follow-up.

Average (mm)	1.47	
Standard deviation (mm)	0.99	
Number (%) of implants with readable radiographs	107 (65.2%)	
Frequencies	*n*	%
0 mm	0	0
0.1–1 mm	45	42.1
1.1–2 mm	47	43.9
2.1–3 mm	13	12.1
>3 mm	2	1.9

**Table 6 jcm-13-01549-t006:** Biological complications for short implants during the study follow-up.

Number	Implant Position	Follow-Up in Months	Biological Complication	Treatment ^1,2^	Outcome
1	36	4	Probing pocket depth of 5 mm	Non-surgical	Unresolved
	37	2	Probing pocket depth of 5 mm		
2	36	43	Peri-implant pathology, suppuration	Antibiotic and surgical	Unresolved
	37	47	Peri-implant pathology		
3	45	23	Peri-implant pathology	Non-surgical	Resolved
4	36	84	Peri-implant pathology	Non-surgical	Implant failure
5	47	48	Peri-implant pathology	Non-surgical	Resolved
6	16	35	Probing pocket depth 6 mm; suppuration	Non-surgical	Unresolved
7	35	30	Peri-implant pathology	Non-surgical	Resolved
8	36	2	Suppuration	Non-surgical	Resolved
	37	2		Antibiotic	Resolved
9	45	58	Peri-implant pathology	Non-surgical	Unresolved
	46	58		Non-surgical	Resolved
14	37	1	Suppuration	Non-surgical	Resolved
10	45	18	Peri-implant pathology	Non-surgical	Resolved
	46	18		Non-surgical	Resolved
17	16	72	Peri-implant pathology	Non-surgical	Resolved
11	34	59	Peri-implant pathology	Non-surgical	Resolved
	35	76	Peri-implant pathology; suppuration	Non-surgical	Unresolved
	36	59	Peri-implant pathology	Non-surgical	Unresolved
12	46	15	Peri-implant pathology	Non-surgical	Resolved
13	46	4	Probing pocket depth 5 mm	Non-surgical	Implant failure
14	47	71	Suppuration	Non-surgical	Unresolved
				Antibiotic	

^1^—non-surgical treatment involving removal of bacterial plaque and irrigation with 0.2% chlorhexidine gel; ^2^—surgical treatment involving removal of granulation tissue and decontamination of the implant surface with 0.2% chlorhexidine, together with the administration of antibiotics.

**Table 7 jcm-13-01549-t007:** Mechanical complications for short implants during the study follow-up.

Patient Number	Implant Position	Follow-Up in Months	Mechanical Complication	Treatment
1	16	36	Prosthetic screw loosening	Retightening and adjusted occlusion
2	35	64	Prosthetic screw loosening	Retightening and adjusted occlusion
	37			
3	15	1	Abutment screw loosening	Retightening and adjusted occlusion
4	35	1	Abutment screw loosening	Retightening and adjusted occlusion
	36			
5	36	10	Prosthetic screw loosening	Retightening and adjusted occlusion
6	36	8	Abutment screw loosening	Retightening and adjusted occlusion
7	36	1	Abutment screw loosening	Retightening and adjusted occlusion
	37			
8	36	13	Abutment screw loosening	Retightening and adjusted occlusion
9	16	13	Prosthetic screw loosening	Retightening and adjusted occlusion
10	36	1	Abutment screw loosening	Retightening and adjusted occlusion
	46	89	Prosthetic screw loosening	
11	36	73	Definitive crown fracture	Prostheses replacement
12	36	20	Prosthetic screw loosening	Retightening and adjusted occlusion
	46	48	Provisional crown fracture	Mended in the laboratory
13	46	10	Prosthetic screw loosening	Retightening and adjusted occlusion
14	16	31	Provisional crown fracture	Prostheses replacement
15	36	80	Definitive crown fracture	Prostheses replacement
16	44	2	Abutment screw loosening	Retightening and adjusted occlusion
17	46	7	Provisional crown fracture	Mended in the laboratory
	47			
18	26	16	Prosthetic screw loosening	Retightening and adjusted occlusion
19	47	31	Prosthetic screw loosening	Retightening and adjusted occlusion
20	45	1	Abutment screw loosening	Retightening and adjusted occlusion
21	35	2	Abutment screw loosening	Retightening and adjusted occlusion
	45			
22	35	2	Abutment screw loosening	Retightening and adjusted occlusion
23	27	1	Abutment screw loosening	Retightening and adjusted occlusion
24	46	4	Abutment screw loosening	Retightening and adjusted occlusion
25	46	15	Prosthetic screw loosening	Retightening and adjusted occlusion
26	45	24	Provisional crown fracture	Prostheses replacement
	46			
27	14	1	Prosthetic screw loosening	Retightening and adjusted occlusion

## Data Availability

Data access will be provide by the authors upon reasonable request.

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
