# Peer review of "Partial Implant Rehabilitations in the Posterior Regions of the Jaws Supported by Short Dental Implants (7.0 mm): A 7-Year Clinical and 5-Year Radiographical Prospective Study"

_jcm, 2024, doi:10.3390/jcm13061549_

Round 1

Reviewer 1 Report

Comments and Suggestions for Authors

The manuscript titled “Partial implant rehabilitations in the posterior regions of the jaws supported by short dental implants (7.0 mm): A 7-years clinical and 5-years radiographical study” has been submitted to the Journal of Clinical Medicine.

This study aimed to explore the outcomes of short implants inserted in the posterior maxilla or mandible using one-stage or immediate function protocols, with a follow-up of 7 years (clinical) and 5 years (radiographic). The findings suggest that utilizing 7 mm long implants for partially implant-supported rehabilitation of atrophic posterior jaws is feasible in the long term. However, stringent case selection is crucial, particularly for smokers and when employing implant configurations that ensure a minimum of one unit in inter-implant distance.

While the manuscript addresses a compelling issue, there are several concerns related to the study.

Title: Even when stating that it is a prospective study, the type of study should be included.

Abstract

- The numbers corresponding to the subsections should be removed.

- No background is presented before the objective.

- In lines 26 and 27, it seems that the mean and standard deviation are presented. Please review.

- The objective mentions clinical and radiographic follow-up. Similarly, results and conclusions regarding these aspects should be presented.

Keywords: Please ensure that all of them correspond to MeSH terms.

Introduction

- The information described in line 47 should be moved to the study methodology.

- In line 59, the referencing format should be reviewed.

- The STROBE guidelines recommend presenting a hypothesis.

Methods

- It appears that this study aligns more with a follow-up study to a cohort. Typically, cohort studies have a comparison group, which is lacking in this study. Considering that this study followed the parameters of the STROBE guidelines, it is recommended that the revised version includes the STROBE checklist as an attached file, indicating the line and page where each criterion is met.

- Please provide the city and country to which the Health Ethics Committee belongs. The Declaration of Helsinki was introduced in 1964, with its latest amendment in 2013. Please review.

- The information presented between lines 88 and 120 pertains to the results.

- The level of calibration of the clinicians who assessed the results during the follow-up and the statistical test used should be presented.

- Figures 3 to 5 belong to the results section.

Results

Please consider the information that needs to be transferred to this section.

Discussion

- Please provide scientific support for the content described between lines 316 and 321.

- The study has other limitations, such as the study type, and others that were mentioned above.

Conclusions

The objective mentions clinical and radiographic follow-up. Similarly, conclusions regarding these aspects should be presented.

It is recommended to review the references and use those with the highest available evidence. Some of them are more than 20 years old.

A detailed review of the language and grammar is necessary.

Comments on the Quality of English Language

Moderate editing

Author Response

Reviewer 1

Comments and Suggestions for Authors

The manuscript titled “Partial implant rehabilitations in the posterior regions of the jaws supported by short dental implants (7.0 mm): A 7-years clinical and 5-years radiographical study” has been submitted to the Journal of Clinical Medicine.

This study aimed to explore the outcomes of short implants inserted in the posterior maxilla or mandible using one-stage or immediate function protocols, with a follow-up of 7 years (clinical) and 5 years (radiographic). The findings suggest that utilizing 7 mm long implants for partially implant-supported rehabilitation of atrophic posterior jaws is feasible in the long term. However, stringent case selection is crucial, particularly for smokers and when employing implant configurations that ensure a minimum of one unit in inter-implant distance.

While the manuscript addresses a compelling issue, there are several concerns related to the study.

  1. Title: Even when stating that it is a prospective study, the type of study should be included.

Response: The authors thank the Reviewer’s suggestion. The title was adjusted as requested.

Changes: Title: “Partial implant rehabilitations in the posterior regions of the jaws supported by short dental implants (7.0 mm): A 7-years clinical and 5-years radiographical prospective study”

Abstract

  1. The numbers corresponding to the subsections should be removed.

Response: The authors thank the Reviewer for the indication. Sections numbers were removed as requested.

Changes: lines 16,20,26,33

  1. No background is presented before the objective.

Response: The authors thank the Reviewer for the indication. A background was added as requested.

Changes: lines 16-18

  1. In lines 26 and 27, it seems that the mean and standard deviation are presented. Please review.

Response: The authors thank the Reviewer for the indication. The statistical analysis was indicated on the Materials and Methods section for clarity.

Changes: lines 25,26

  1. The objective mentions clinical and radiographic follow-up. Similarly, results and conclusions regarding these aspects should be presented.

Response: The authors thank the Reviewer for the suggestion. The authors complemented the results section and incorporated the survival and marginal bone loss results on the conclusion section as requested.

Changes: lines 30,35,36

  1. Keywords: Please ensure that all of them correspond to MeSH terms.

Response: The authors thank the Reviewer for the indication. MeSH terms were corrected as requested: Terms removed: Short implants, immediate function, Oxidized surface; Terms introduced:  dental implants, immediate loading, edentulous jaws.

Changes: Line 39.

Introduction

  1. The information described in line 47 should be moved to the study methodology.

Response: The authors thank the Reviewer’s suggestion. The information was moved to the Methods section as requested.

Changes: Lines 86-87, 96

  1. In line 59, the referencing format should be reviewed.

Response: The authors thank the Reviewers indication. The referencing format was adjusted as requested: [9,10,14,15,16,17,18] à[9,10,14-18]

Changes: Line 66

  1. The STROBE guidelines recommend presenting a hypothesis.

Response: The authors thank the Reviewers recommendation. A hypothesis was included as recommended.

Changes: Lines 81 and 82

Methods

  1. It appears that this study aligns more with a follow-up study to a cohort. Typically, cohort studies have a comparison group, which is lacking in this study.

Response: The authors thank the Reviewer’s indication. The study design was replaced from “cohort” to “case series” for clarity.

Changes: Line 86

  1. Considering that this study followed the parameters of the STROBE guidelines, it is recommended that the revised version includes the STROBE checklist as an attached file, indicating the line and page where each criterion is met.

Response: The authors thank the Reviewer for the recommendation. The checklist was included.

Changes: STROBE checklist file

  1. Please provide the city and country to which the Health Ethics Committee belongs. The Declaration of Helsinki was introduced in 1964, with its latest amendment in 2013. Please review.

Response: The authors thank the Reviewer’s query. City and country of the Ethics Committee, and the introduction and amended dates of the Helsinki Declaration were included.

Changes: Lines 90 and 91.

  1. The information presented between lines 88 and 120 pertains to the results.

Response: The authors thank the Reviewers suggestion. The information was moved to the Results section.

Changes: Lines 166-183 and 191-199

  1. The level of calibration of the clinicians who assessed the results during the follow-up and the statistical test used should be presented.

Response: The authors thank the Reviewer’s query. The overall inter-examiner reliability was estimated using a weighted average of the pairwise inter-examiner reliability estimates. Inter-examiner reliability results for weighted kappa scores

were 0.83.

Changes: Lines 160 to 163

  1. Figures 3 to 5 belong to the results section.

Response: The authors thank the Reviewer’s indication. The figures were moved to the results section as requested.

Changes: Lines 185-190

Results

  1. Please consider the information that needs to be transferred to this section.

Response: The authors thank the Reviewers indication. All information from points 13 and 15 were moved as requested.

Changes: Lines 160 to 199.

Discussion

  1. Please provide scientific support for the content described between lines 316 and 321.

Response: The authors thank the reviewer’s request. Actually this result was from our study, and in the following phrases we provided discussion and scientific support. The authors made it clear on the manuscript.

Changes: Line 356

  1. The study has other limitations, such as the study type, and others that were mentioned above.

Response: The authors thank the Reviewer’s indications. The limitations list was amended as suggested.

Changes: Line 383

Conclusions

  1. The objective mentions clinical and radiographic follow-up. Similarly, conclusions regarding these aspects should be presented.

Response: The authors thank the Reviewer’s recommendation. The conclusions were adjusted as requested.

Changes: Lines 392-394

  1. It is recommended to review the references and use those with the highest available evidence. Some of them are more than 20 years old.

Response: The authors thank the Reviewer’s indication. It is a fact that some references are > 20 years old, nevertheless it is the case due to the lack of studies on this particular subject. Considering the references >20 years:

Ref 1 and 2 – Necessary to point out the techniques used in the beginning of the millenium to establish the background.

Ref 15 – Important to refer that the cumulative survival rate of short implants was low in a previous study

Ref 25 – Important because of the follow-up similar to our study

Ref 26 – Important because it evaluates the effect of inter-implant distances, an important factor in our study that showed significant differences in implant survival

Ref 32 and 33 – Both these references placed implants in augmented bone, an alternative to using short implants, with a similar follow-up to our study, therefore necessary in the discussion of results.

Ref 35 – In this study, implants were placed after alveolar nerve transposition, an alternative to using short implants, therefore they are necessary in the discussion of results.

Changes: None

  1. A detailed review of the language and grammar is necessary.

Response: The authors thank the Reviewer’s indication. Proof-read and corrected.

Changes: Throughout the manuscript.

Comments on the Quality of English Language

  1. Moderate editing

Response: The authors thank the Reviewer’s indication. Proof-read and corrected.

Changes: Throughout the manuscript.

Reviewer 2 Report

Comments and Suggestions for Authors

Dear Authors , your paper is a revision with prolonged follow-up of an already published paper but with a short follow-up. Considering that the follow-up is the main aspect in such case studies I maintain that the paper should be evaluated by referres but after the EditorinChief decision about the possibility to publish it after revision. I'm sorry for such delay but sincerely I don't know what to do in such instance. I hope to evaluate it soon. Kind regards

Author Response

Dear Authors , your paper is a revision with prolonged follow-up of an already published paper but with a short follow-up. Considering that the follow-up is the main aspect in such case studies I maintain that the paper should be evaluated by referres but after the EditorinChief decision about the possibility to publish it after revision. I'm sorry for such delay but sincerely I don't know what to do in such instance. I hope to evaluate it soon. Kind regards

Response: The authors thank the Reviewer's comment. The manuscript received the go ahead by the Editor and was marked as major revisions. The authors proceeded to perform significant revisions and the manuscript is now stronger. We hope you find the revisions suitable and we remain at your disposal should you judge that more revisions are necessary.

Changes: Throughout the manuscript.

Reviewer 3 Report

Comments and Suggestions for Authors

It would be advisable to describe whether patients needed connective grafts to reduce soft tissue complications that usually occur in patients with bone atrophy.

It would be advisable for the authors to search for literature with results similar to those of their sample that could explain the lower survival of implants in the maxilla.

It would be advisable for the authors to discuss treatment alternatives in the atrophic maxilla and the survival results of different studies (standard implants with open or closed sinus elevation with or without regeneration).

Is the use of short implants the treatment of choice used by your group for the rehabilitation of the atrophic posterior maxillary sector or do you use another alternative treatment?

Author Response

Comments and Suggestions for Authors

  1. It would be advisable to describe whether patients needed connective grafts to reduce soft tissue complications that usually occur in patients with bone atrophy.

Response: The authors tank the Reviewer’s query. In this study no patients had connective tissue grafts provided. Only the implant rehabilitation was provided with incisions performed on the crest’s palatal aspect with parallel discharges towards the vestibulum to allow maximum tissue repositioning of the vestibular aspect of the flap. The authors clarified the manuscript.

Changes: Line 109.

  1. It would be advisable for the authors to search for literature with results similar to those of their sample that could explain the lower survival of implants in the maxilla.

Response: The authors thank the Reviewer’s query. The potential reasons for these results were introduced as requested.

Changes: Lines 272-275, refence no.27

  1. It would be advisable for the authors to discuss treatment alternatives in the atrophic maxilla and the survival results of different studies (standard implants with open or closed sinus elevation with or without regeneration).

Response: The authors thank the Reviewer’s query. Treatment alternatives were considered in the discussion as suggested. However, only 2 studies were introduced given the similar follow-up compared to the present study.

Changes: Lines 298-299, 310-313; References 37 and 38

  1. Is the use of short implants the treatment of choice used by your group for the rehabilitation of the atrophic posterior maxillary sector or do you use another alternative treatment?

Response: The authors thank the Reviewer’s query. The use of short implants during the follow-up of this particular study was the treatment of choice for the rehabilitation of the atrophic posterior maxillary sector.

Changes: None.

Reviewer 4 Report

Comments and Suggestions for Authors

Dear Authors,

Introduction

I suggest expanding the introduction by explaining the advantages and disadvantages of short implants. 

Line 59 - briefly explain why these studies gave different results.

Line 67 - write the objective in more detail, introducing the outcomes that were evaluated and the null hypothesis. 

Materials and methods

The materials and methods were thoroughly and adequately written; however, I strongly recommend to break it down into paragraphs so as to distinguish the procedure from the clinical outcomes.

Results

The results are unclearly written.

I suggest the subdivision of the paragraphs as follows:

1. Sample selection

2. Implants survival rate

3. Marginal bone loss

4. Biological complications

5. Mechanical complications

For each category, additional tables should be included in order to schematise the results and make the paper easier to read. 

Comments on the Quality of English Language

English quality is good.

Author Response

Comments and Suggestions for Authors

Dear Authors,

Introduction

  1. I suggest expanding the introduction by explaining the advantages and disadvantages of short implants.

Response: The authors thank the Reviewer’s indication. A paragraph explaining the advantages and disadvantages of short implants was introduced as requested.

Changes: Lines 47-52.

  1. Line 59 – briefly explain why these studies gave different results.

Response: The authors thank the Reviewer’s query. The results were due to different study designs, sample characteristics and follow-up. A brief explanation was introduced as requested.

Changes: Line 67

  1. Line 67 – write the objective in more detail, introducing the outcomes that were evaluated and the null hypothesis.

Response: The authors thank the Reviewer’s suggestion. The objective was written in more detail with the introduction of the outcomes that were evaluated together with a working hypothesis (given inferential statistics were disabled due to study design and sample characteristics).

Changes: Lines 79-82

Materials and methods

  1. The materials and methods were thoroughly and adequately written; however, I strongly recommend to break it down into paragraphs so as to distinguish the procedure from the clinical outcomes.

Response: The authors thank the Reviewer’s indication. The materials and methods were broken down into paragraphs separating the procedure from the clinical outcomes. Now it reads a paragraph for implant insertion, another for abutment selection and prosthetic protocols.

Changes: Lines 104-124

Results

  1. The results are unclearly written.

I suggest the subdivision of the paragraphs as follows:

-Sample selection

-Implants survival rate

-Marginal bone loss

-Biological complications

-Mechanical complications

Response: The authors thank the Reviewer’s indication. The subdivision of paragraphs was performed as requested.

Changes: Lines 165, 178, 200, 222, 228, 236

  1. For each category, additional tables should be included in order to schematise the results and make the paper easier to read.

Response: The authors thank the Reviewers indication. New tables for each category were introduced as requested.

Changes: Tables 5, 6 and 7         

  1. Comments on the Quality of English Language:

English quality is good.

Response: The authors thank the Reviewer for the constructive review. Changes in English were performed at the request of other reviewers.

Changes: Throughout the manuscript

Reviewer 5 Report

Comments and Suggestions for Authors

Dear Authors,

I carefully read your manuscript entitled: "Partial implant rehabilitations in the posterior regions of the 2 jaws supported by short dental implants (7.0 mm): A 7-years 3 clinical and 5-years radiographical study."
Please, follow the suggestions below.

Introduction

This part needs to be better developed. You aggregate several citations, please explain better each key message you want to point out.

Additionally, regarding short implants, it has to be mentioned that the protocol used makes the difference, predictability, patient compliance and implant survival. In light of this, please add these references:

https://pubmed.ncbi.nlm.nih.gov/33374157/

https://pubmed.ncbi.nlm.nih.gov/27389435/

Discussion In this part you have to analyze the impact of the width of the keratinized mucosa on the implant survival and the new techniques proposed in literature.

Comments on the Quality of English Language

Minor editing of English language required

Author Response

Comments and Suggestions for Authors

Dear Authors,

I carefully read your manuscript entitled: “Partial implant rehabilitations in the posterior regions of the 2 jaws supported by short dental implants (7.0 mm): A 7-years 3 clinical and 5-years radiographical study.”

Please, follow the suggestions below.

Introduction

  1. This part needs to be better developed. You aggregate several citations, please explain better each key message you want to point out.

Response: The authors thank the Reviewer’s suggestion. The text was clarified by separating into different paragraphs according to the message.

Changes: Lines 47-52,66,72-75.  

  1. Additionally, regarding short implants, it has to be mentioned that the protocol used makes the difference, predictability, patient compliance and implant survival. In light of this, please add these references: https://pubmed.ncbi.nlm.nih.gov/33374157/; https://pubmed.ncbi.nlm.nih.gov/27389435/

Response: The authors thank the Reviewer’s suggestion. The phrase was introduced in the manuscript for clarity along with the suggested references.

Changes: Line 72-75, references nº 19 and 20.

Discussion

  1. In this part you have to analyze the impact of the width of the keratinized mucosa on the implant survival and the new techniques proposed in literature.

Response: The authors thank the Reviewer’s suggestion. A paragraph was introduced in the Discussion section discussing the impact of keratinized mucosa on the outcome as well as the new techniques proposed in recent literature.

Changes: Lines 331-346, references 39, 40, 41, 42, 43, 44, 45 and 46.

  1. Comments on the Quality of English Language

Minor editing of English language required

Response: The authors thank the Reviewer for the constructive review. English editing was performed as requested.

Changes: Throughout the manuscript.

Round 2

Reviewer 1 Report

Comments and Suggestions for Authors

The authors have improved the quality of the manuscript significantly. Its publication is recommended.

Comments on the Quality of English Language

minor

Reviewer 4 Report

Comments and Suggestions for Authors

Corrections were relevant and appropriate to what was required.